# Stepwise on-surface synthesis of nitrogen-doped porous carbon nanoribbons
Jin Xu[1], Shuaipeng Xing[1], Jun Hu [2]✉ & Ziliang Shi [1]✉

Precise synthesis of carbon-based nanostructures with well-defined structural and chemical properties is of significance towards organic nanomaterials, but remains challenging. Herein, we report on a synthesis of nitrogen-doped porous carbon nanoribbons through a stepwise on-surface polymerization. Scanning tunneling microscopy revealed that the selectivity in molecular conformation, intermolecular debrominative aryl-aryl coupling and inter-chain dehydrogenative cross-coupling determined the well-defined topology and chemistry of the final products. Density functional theory calculations predict that the ribbons are semiconductors, and the band gap can be tuned by the width of the ribbons.

Carbon-based organic nanoarchitectures have attracted particular interest in past decades, for their potential applications in catalysis, nanoelectronics and optoelectronic devices[1–4]. The distinct properties of these organic nanoarchitectures are determined essentially by the structural and chemical characteristics that can be customized with atomic precision[5–11]. Tremendous effort based on bottom-up strategy has been made to tune the band structure of these nanomaterials, which range from one-dimensional (1D) π-conjugated polymers, graphene-based nanoribbons to two-dimensional (2D) polyarylene networks and porous graphenes[12–26]. To name a few, the judicious choice of molecular precursors allows for the synthesis of graphene nanoribbons with different widths, edge-states or doping[12–22]. In particular, nitrogen-doping can adjust the electronic properties of graphene nanoribbons[18,27]. Recent studies have demonstrated nanopatterns in graphenes or in nanoribbons rendering new band structures[8,10,11,18,28,29]. Theoretical calculations have predicted novel spintronic properties for the porous polyarylene networks and nanoribbons[30–32]. Nevertheless, a precise synthesis of carbon-based nanoarchitectures with well-defined topology and chemistry is a huge challenge, because it needs a delicate control on both the molecular precursors conformation/composition and the covalent coupling reaction routes.

In this report, we demonstrate a stepwise on-surface polymerization towards a nitrogen-doped porous carbon nanoribbon (*n*-NPCN) structure with various widths, *n*. The precursor molecule 6,6"-dibromo-2,2':6',2"-terpyridine (DT) (Fig. 1a) contains a terpyridine group and two Br terminals. The molecule has a flexible conformation, as its bipyridine moieties can adopt *cis*- or *trans*-configuration[33]. Scanning tunneling microscopy (STM) revealed that the nanoribbons were precisely synthesized owing to

the intrinsic selectivity both in the molecular adsorption and in a two-step reaction on Au(111) (see Section Methods). As shown in Fig. 1a, an initial debrominative aryl-aryl coupling (after 360–510 K annealing), between the molecular *trans*-conformers, creates zig-zag polymeric chains (ZZCs) (Fig. 1b). The selective lateral fusion of ZZCs (and as-formed nanoribbons) after an inter-chain dehydrogenation at 635–775 K annealing, leads to the formation of *n*-NPCNs (Fig. 1c, d). Control experiments performed on Ag(111) and Cu(111) indicated an important role of the substrate to the structural selectivity of NPCNs. Density function theory (DFT) calculation (see Section Methods) on the band structure of the ribbons suggests a band gap ~3.2 eV depending on the width of the ribbons.

## Results and discussion

### Intermolecular debrominative aryl-aryl coupling of DT on Au(111)

DT molecules maintain intact when absorbing on a Au(111) surface held at 293 K (room temperature). As shown in Fig. 2a, STM observation of the sample at 113 K revealed the molecular monomers adopting the *trans*-configuration with Br atoms at the ends (Fig. 2b), and aggregating into arrays due to intermolecular halogen bonding and −C–H···Br hydrogen bonding. (The distance between the adjacent Br measures 3.4 Å; the distance between Br and the closest peripheral H atoms belonging the surrounding DTs measures 3.3 Å. Both distances fall in the typical length range of the halogen and hydrogen bonds, respectively[34–36].) The coverage of DT was 0.54 ML (monolayer; 1 ML is defined as an entire coverage of the close-packed structure). The predominance of the *trans*-conformers results from their lower free energy and the low energy barrier of *trans*-*cis* transition[33]. Such a conformational preference furthermore leads to a topological

[1]Center for Soft Condensed Matter Physics & Interdisciplinary Research, School of Physical Science and Technology, Soochow University, Suzhou 215006, China. [2]School of Physical Science and Technology, Ningbo University, Ningbo 315112, China. ✉e-mail: hujun2@nbu.edu.cn; phzshi@suda.edu.cn

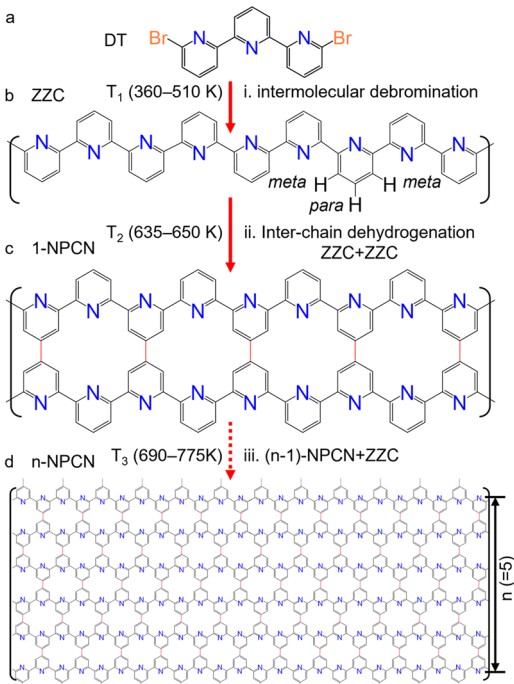

**Fig. 1 | Stepwise synthesis of nitrogen-doped porous carbon nanoribbons (NPCNs) on Au(111). a** Chemical structure of the molecular precursor 6,6"-dibromo-2,2':6',2"-terpyridine (DT). **b** Zigzag polymeric chains (ZZCs) through (i) intermolecular debrominative aryl-aryl coupling at 360–510 K annealing. **c** 1-NPCN through a lateral fusion of two ZZCs after (ii) an inter-chain dehydrogenation at 635–650 K annealing. **d** n-NPCN through (iii) a lateral fusion of a ZZC polymer and an (n − 1)-NPCN after 690–775 K annealing.

selectivity of the resulting polymers after intermolecular covalent coupling (*vide infra*).

After annealing to 360 K, long zigzag chains (ZZCs) emerged (Fig. 2c), and assembled into large array domains. These ZZC polymers, measuring a periodicity of 7.2 Å (Fig. 2d), result from a covalent coupling between molecular precursors after a debrominative reaction. This observation agrees with a recent report[37]. The cleaved Br atoms ('×' in Fig. 2d) stay in between chains and assist their packing through Br-H bonds. The length of Br···H bonds measures 3.3–4.4 Å, in accordance with previous reports[35,36]. Frequently, long ZZCs extend over entire terraces without foldings or kinks. Our statistical analysis (see Supplementary Fig. S1 in Supporting Information (SI)) on the length of ZZCs gives an average length ~ 31 nm. We note that the intermolecular covalent coupling prefers a *trans*-mode, because a *cis*-mode coupling otherwise may bring out large steric repulsion between the peripheral H atoms[17,37–40].

## Formation of *n*-NPCNs

Next, we annealed the sample to a high temperature (635–775 K), to activate the lateral fusion of ZZCs via an inter-chain dehydrogenation reaction. Fig. 3a depicts an STM overview of the sample after an annealing at 650 K for 60 min. Ribbon structures, having regularly aligned pores and zigzag edges, appeared. Most of ribbons have a width of one-row-of-pores, namely 1-NPCN (Fig. 3b). A typical 1-NPCN measures an inter-chain distance of ~6.7 Å, and exhibits a homogeneous contrast, which suggests a covalent inter-chain coupling. Thus, the 1-NPCN from a lateral fusion of two ZZCs results from an inter-chain dehydrogenation reaction[10,14,16,41]. We note that at the ends of 1-NPCNs, branched ZZCs grow out from the ribbon, which suggests a 'zipper' mechanism[42]. Our schematic model (bottom, Fig. 3c) illustrates that the inter-chain coupling occurs via a selective dehydrogenation reaction between two para-H atoms. This selectivity can be rationalized by the huge steric hindrance between otherwise meta-H atoms. The 'zipper'-like growth also takes place between 1-NPCNs and ZZCs, leading to the formation of 2-NPCNs. Distinct from the growth of 1-NPCN

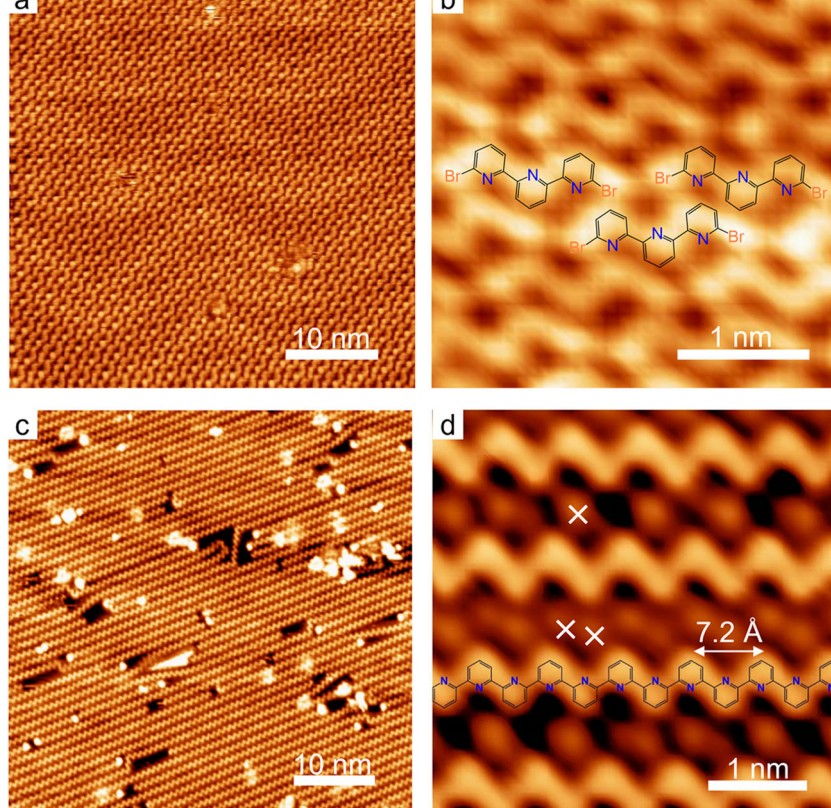

**Fig. 2 | Intermolecular debrominative aryl-aryl coupling of DT on Au(111). a** STM overview of the close-packing assembly of intact DT monomers. **b** High-resolution STM of the close-packing. The three molecular models indicate the intermolecular halogen interactions. **c** Polymeric zigzag chains (ZZCs) after intermolecular debrominative coupling at a 360 K annealing. **d** High-resolution STM of ZZCs. Symbols '×' mark the cleaved Br atoms.

**Fig. 3 | Formation of *n*-NPCNs. a** STM overview of the sample after an annealing at 650 K for 60 min. Two ovals indicate the bending or vibration of ZZCs. **b** STM of a 1-NPCN. **c** STM of a 1-NPCN with branched ZZC at the end (upper) and an illustration (lower) depicting the 'zipper' mechanism. **d** STM of a 2-NPCN and the branched ZZC (upper). The 'zipper' mechanism in the formation of 2-NPCNs (lower). **e** STM overview of the sample after an annealing at 745 K for 60 min. **f** STM images of 3-, 4- and 6-NPCN. Color in structural models: C, grey; N, blue; H, white. Scale bars in **b**–**d** and **f**: 2 nm.

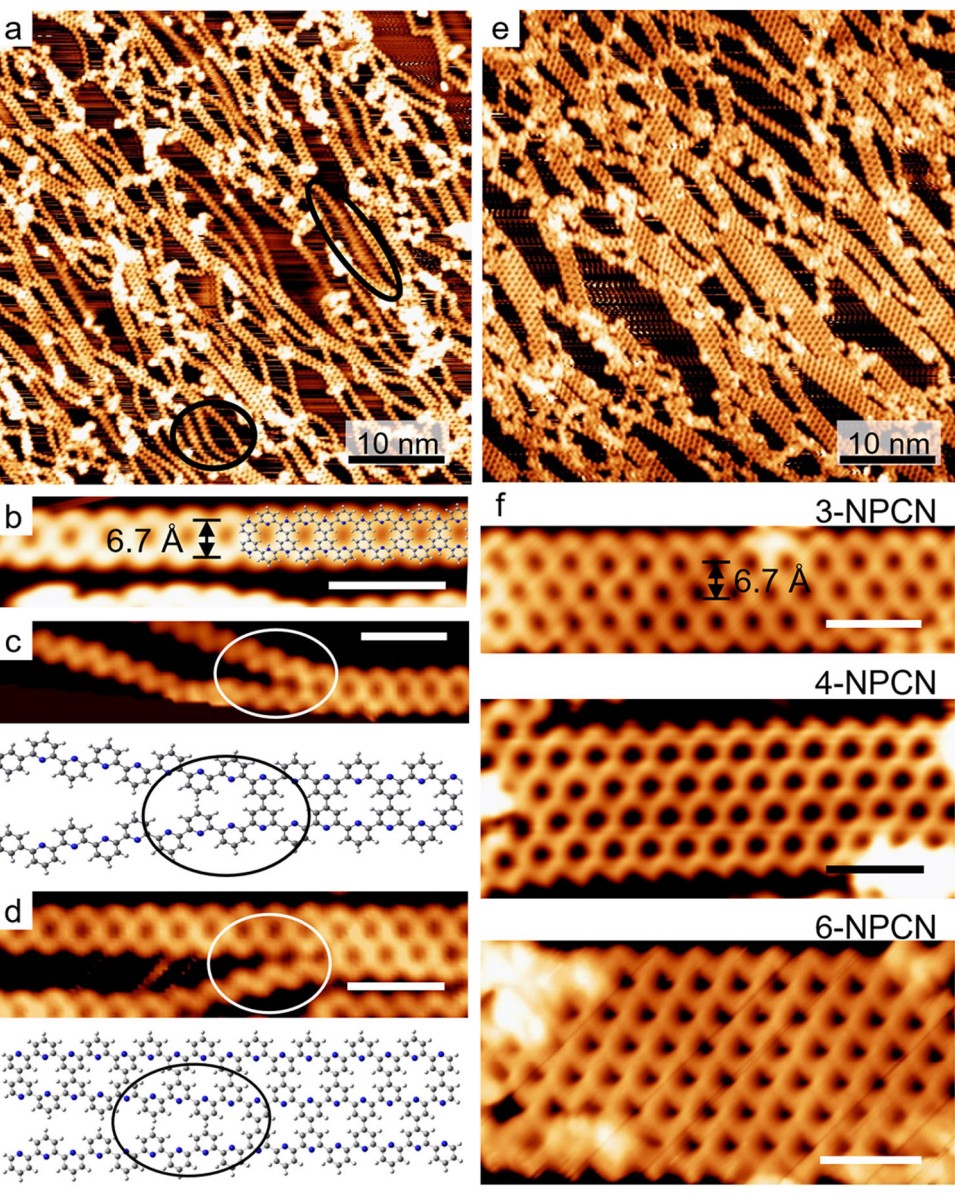

—a fusion of two ZZCs, the growth of 2-NPCNs is based on the fusion of one ZZC and an as-formed 1-NPCN, ZZC + 1-NPCN → 2-NPCN; see Fig. 3d. We note that ZZCs have flexible backbones, as suggested by their bending shape or the vibrational motion shown in our STM scanning at room temperature (see ovals in Fig. 3a and Supplementary Fig. S2 in SI). In contrast, the ribbon has a rigid and straight backbone. These observations imply that wider ribbons can form only in an 'accumulative' manner; that is, the fusion of a ZZC to an as-formed ribbon, ZZC + (*n* − 1)-NPCN → *n*-NPCN (Fig. 3d). To this end, we have annealed the sample at a higher temperature (745 K) for 60 min. There occurred a considerable increment of the number and the width of ribbon structures at the cost of ZZCs (Fig. 3e). The resulting *n*-NPCNs have various widths; *n* ranges from 1 to 6. Figure 3f displays the STM topography of typical *n*-NPCNs. The neighboring ZZC components within a ribbon measures a distance of ~6.7 Å (top panel, Fig. 3f), indicating that all ribbons are formed through inter-chain covalent coupling after the selective dehydrogenation.

## The catalytic property of different substrates to the selection effects

As aforementioned, the intrinsic selectivity in the molecular conformation, debrominative reaction and dehydrogenation have played important roles in the formation of *n*-NPCNs. To test the catalytic property of different substrates to the selection effects, we have carried out control experiments on Ag(111) and Cu(111) surfaces. After an annealing at 520 K to a Ag(111) sample pre-deposited with 0.38 ML molecules, similar ZZC polymers appeared (Fig. 4a); see Fig. S1 for the length distribution. After an annealing at 640 K for 60 min, the polymers began to fuse. As indicated in Fig. 4b, a typical STM overview displays short 1-NPCNs (the rectangle) appearing via a lateral fusion of two ZZCs. Such an onset temperature is quite close to the activation temperature (635–650 K) for the dehydrogenation on Au(111) (*vide supra*), while it is much higher than that for the Cu(111) samples (*vide infra*). The trend in the activation temperatures on different surfaces reflects the catalytic property of substrates that agrees with the similar trend documented in on-surface dehalogenative reactions. Next, an annealing at 690 K for 60 min led to the emergence of wider *n*-NPCNs (Fig. 4c). The rectangle in Fig. 4c denotes a 4-NPCN.

On Cu(111), molecules (0.50 ML) formed ZZCs after an annealing at 360 K, which however contained more foldings or kinks, and thus had shorter length (Fig. 4d); see Supplementary Fig. S1 for the length distribution. A magnified STM image (Fig. 4e) depicts the foldings, kinks or even single rings (see the structural model in the inset), which result from the coupling of *cis*-conformers or from the *cis*-mode coupling[17,39,40]. These non-

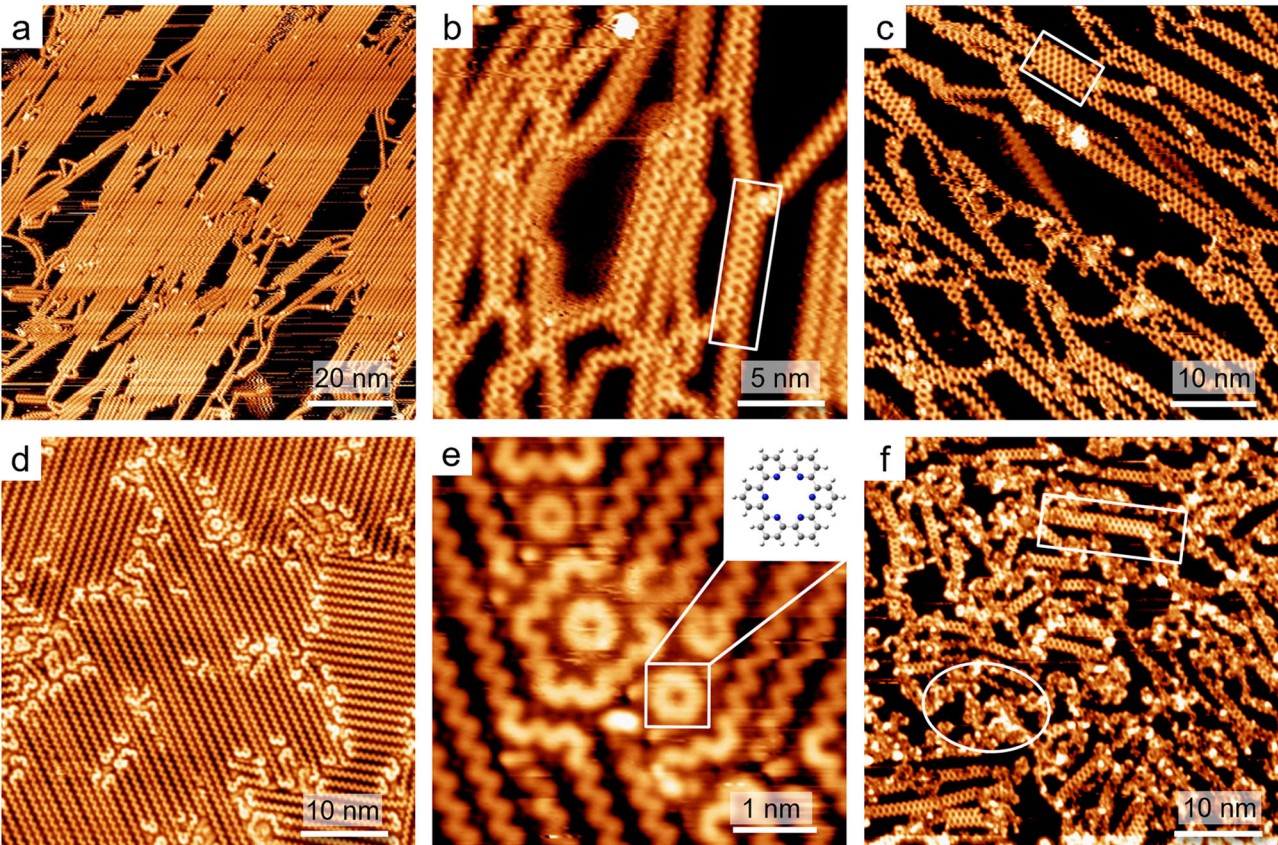

**Fig. 4 | Control experiments on Ag(111) and Cu(111). a** ZZCs on Ag(111) after a 520 K-annealing. **b** 1-NPCNs (white rectangle) emerging after a 640 K-annealing. **c** n-NPCNs after a 690 K-annealing. **d** ZZCs on Cu(111) after a 360 K-annealing. **e** A close inspection of non-chain products. The inset illustrates a ring from the cyclization of two DTs. **f** n-NPCNs after a 540 K-annealing. The rectangle denotes a 2-NPCN. The oval marks undetermined species.

chain products can be attributed to metal-coordination between Cu adatoms with pyridine ligands[40,43,44]. Our control experiments by adding DT molecules to a Au(111) surface preadsorbed with Cu atoms (see additional STM data in Supplementary Fig. S3 in SI) have also demonstrated the coordination effect leading to the formation of non-chain products[40]. The 540 K-annealing for 60 min of the sample generated n-NPCNs (e.g., a 2-NPCN denoted by the rectangle in Fig. 4f), which however were much narrower and shorter than those on Ag(111) and Au(111) (see Supplementary Fig. S4 in SI), and coexisted with many undetermined species (bright clusters enclosed by the oval in Fig. 4f). According to our proposed scenario of dehydrogenative fusion, ZZC + (n − 1)-NPCN → n-NPCN, in which the size of the reaction precursors (ZZCs and (n − 1)-NPCNs) determines the topology of products (n-NPCNs), the width (and the length) limit of NPCNs on Cu(111) can be attributed to the presence of non-chain products and ZZCs that have limited length. We note that the activation temperatures for both debromination and dehydrogenation reaction on Cu(111) are much lower than that on Au(111) and Ag(111). This difference can be interpreted by the high catalytic reactivity of the Cu surface[45,46].

**The electronic structure of free-standing n-NPCNs**

We have investigated the electronic properties of free-standing n-NPCNs by using DFT calculations. We first depict the atomic and electronic properties of a 2D counterpart of n-NPCNs, the nitrogen-doped polyarylene network (NPN); see the atomic structure of the NPN presented by the inset in Fig. 5b. Distinct from the pure graphene having the $C_6$ rotational symmetry, the lattice of NPN is distorted with two vectors slightly different ($a = 7.14$ Å and $b = 7.46$ Å), and thus the rotational symmetry is reduced to $C_2$. The calculated band structure (Fig. 5a) suggests that the NPN is a semiconductor with an indirect band gap ($E_g$) of about 2.35 eV from the regular GGA-PBE calculation scheme. The valence band maximum (VBM) locates at the M'

point (along **b** of the Brillouin zone) and the conduction band minimum (CBM) at the K point. Note that the electronic energy level at the K' point is only 0.02 eV higher than the CBM. To get more accurate band gap of NPNs, we further used the Heyd–Scuseria–Ernzerhof screened Coulomb hybrid functional (HSE06) to calculate the band structure along the path K→Γ→M', which covers both VBM and CBM, as displayed by the red dashed curves in Fig. 5a. Clearly, the dispersions of the bands are similar to those from the regular GGA-PEB calculation, but $E_g$ increases to 3.15 eV. To further explore the electronic property, we calculated the projected density of states (PDOS) of the p orbitals of the N and C atoms as plotted in Fig. 5b. It can be seen that the PDOS of the p orbitals are mainly separated into two parts, $p_x+p_y$ and $p_z$, and the PDOS of the $p_z$ orbital is quite localized near −3 eV. The $p_x$ and $p_y$ orbitals of the N atom exhibit higher PDOS than those of the C atoms. Moreover, the N atom does not contribute to the VBM, but dominates the CBM. For comparison, we also calculated the band structure of the (free-standing) pure porous graphene (see Supplementary Fig. S5 in SI). Clearly, the pure porous graphene is a direct-band-gap semiconductor, with the band gap of about 2.0 eV with the GGA-PBE calculation scheme, smaller than that for NPNs. In addition, the dispersions of the bands are similar to those for NPNs, but with higher degeneracy due to the higher structural symmetry.

Then we investigated the electronic structure of free-standing n-NPCNs; see additional DFT results in Supplementary Fig. S6. All the n-NPCNs are indirect-band-gap semiconductors with the VBM at the Γ point and the CBM at the X point. Specifically, as shown in Fig. 6a, we plotted the charge densities of the highest occupied molecular orbital (HOMO) and the lowest unoccupied molecular orbital (LUMO) at the Γ and X point for a 1-NPCN. Obviously, both the VBM (the HOMO at the Γ point) and the CBM (LUMO at the X point) are contributed from the π orbitals which originate from the hybridization between the $p_z$ orbitals of the neighboring

C atoms surrounding the hole. Therefore, the electronic states of both the VBM and CBM are localized, resulting in a large band gap. The amplitude of

$E_g$ for $n$-NPCNs increases as the width ($n$) decreases (Fig. 6b), which can be attributed to the effect of the quantum confinement. Particularly, $E_g$ of a ZZC (0-NPCN) is as large as 2.98 eV. $E_g$ of 1-NPCN is 2.53 eV, about 7.6% larger than that for the NPN. For $n$-NPCNs with $n \geq 2$, the band gaps are very close to the NPN, larger by no more than 0.1 eV. Therefore, we conclude that most $n$-NPCNs are also semiconductors and their band gaps are around 3.2 eV, considering the band gap correction from the HSE06 method.

## Conclusion

In conclusion, through the intrinsic selectivity in both the molecular conformation and the two-step reaction routes, we have successfully synthesized a nitrogen-doped porous nanoribbon structure, $n$-NPCN, with well-defined topological and chemical precision. Our DFT calculations suggest that both the nitrogen-doped porous networks and the ribbons are wide band gap semiconductors, with band gaps of ~3.2 eV. We expect that the variable semiconductive properties of $n$-NPCNs may allow for applications in optoelectronic nanodevices.

## Methods

### STM experiments

Sample preparation was performed in an ultrahigh vacuum (UHV) system (SPECS GmbH) at a base pressure ~ $3.0 \times 10^{-10}$ mbar. The single-crystal substrates (Au(111) (Mateck, 99.999%), Ag(111) (Mateck, 99.999%) and Cu (111) (Mateck, 99.999%)) were cleaned by cycles of Ar$^+$ ion sputtering at an energy of 900 eV and annealing at 800–900 K. The molecular precursor 6,6"-dibromo-2,2':6',2"-terpyridine (DT, Aldrich Chemistry, purity >90%) was evaporated from an organic molecular beam epitaxy (DODECON Nanotechnology GmbH), and the sublimation temperature was 210 °C. Cu atoms were evaporated from Cu wires (Puratronic, 99.9999%) contained by a Mo crucible, using a commercial E-beam evaporator. All STM experiments were carried out using an Aarhus SPM apparatus controlled by Nanonis electronics. Topographic data were acquired in the constant current mode, with the bias voltage applied to the sample. The sample temperature at scanning was held at 293 K (room temperature), except for Fig. 2a, b that were acquired with the sample temperature at 113 K.

**Data acquisition conditions of the STM images in figures**. Figure 2 (a) $U = -1.0$ V, $I = 200$ pA; (b) $U = -1.2$ V, $I = 100$ pA; (c) $U = -1.0$ V, $I = 100$ pA; (d) $U = 0.2$ V, $I = 200$ pA. Figure 3 (a) $U = -1.2$ V, $I = 100$ pA; (b) $U = -1.0$ V, $I = 100$ pA; (c) $U = -1.2$ V, $I = 100$ pA; (d) $U = -1.2$ V, $I = 100$ pA; (e) $U = -1.0$ V, $I = 100$ pA; (f) $U = -1.0$ V, $I = 300$ pA;

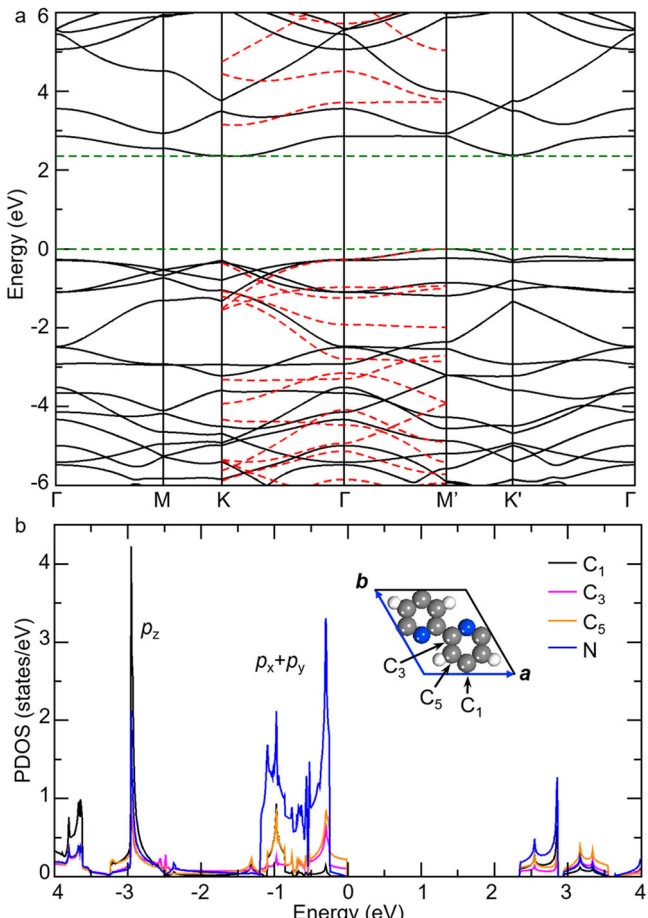

**Fig. 5 | The electronic structure of free-standing nitrogen-doped polyarylene networks (NPNs). a** Band structure. The black and red curves were calculated from the regular GGA-PEB and hybrid HSE06 methods, respectively. The horizontal dashed lines indicate the GGA-PEB band gap. **b** Projected density of states (PDOS) of the N and C atoms. The inset shows the unit cell of the NPN. Color: N, blue; C, grey; H, white.

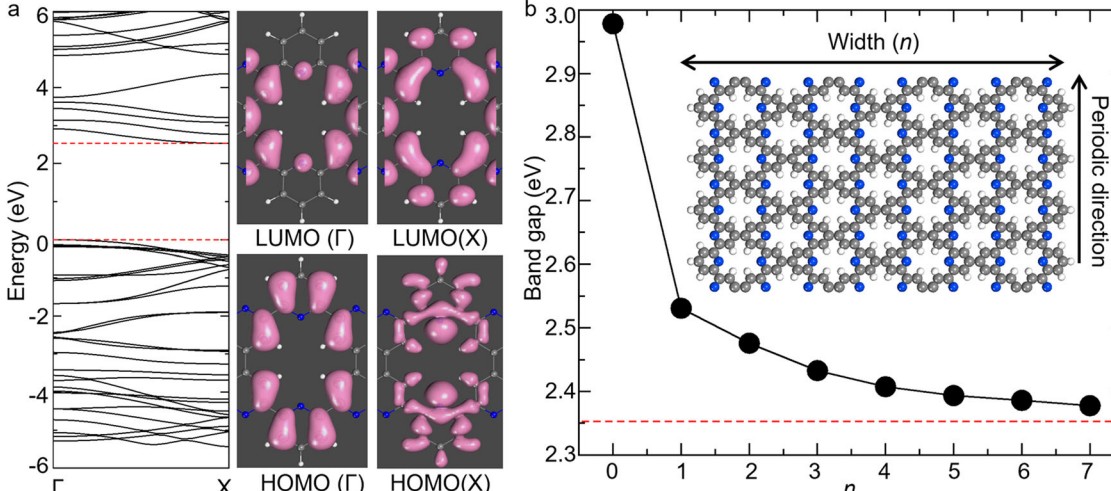

**Fig. 6 | The electronic structure of free-standing $n$-NPCNs. a** Band structure of 1-NPCN, and charge densities of the highest occupied molecular orbital (HOMO) and the lowest unoccupied molecular orbital (LUMO) at the Γ and X point. **b** Band gap of

$n$-NPCNs as a function of the width ($n$). The horizontal dashed line indicates the band gap of NPN. The inset displays the atomic structure of an $n$-NPCN ($n = 7$).

$U = -1.0$ V, $I = 200$ pA; $U = -1.0$ V, $I = 100$ pA. Figure 4 (a) $U = -2.0$ V, $I = 100$ pA; (b) $U = -0.5$ V, $I = 1$ nA; (c) $U = -1.0$ V, $I = 500$ pA; (d) $U = -1.0$ V, $I = 200$ pA; (e) $U = -1.0$ V, $I = 200$ pA; (f) $U = -1.0$ V, $I = 200$ pA.

## DFT calculations

First-principles calculations were carried out by using the Vienna ab-initio simulation package (VASP)[47,48]. The atomic potentials in the framework of the projector augmented wave (PAW) method were adopted[49,50]. The energy cutoff of the expansion of plane waves is set to 500 eV. The generalized gradient approximation (GGA) with the Perdew-Burke-Ernzerhof (PBE) exchange correlation functional was chosen to solve the Schrödinger equation[51]. We also used the Heyd-Scuseria-Ernzerhof screened Coulomb hybrid functional (HSE06) to get more accurate electronic properties, especially the band gaps[52]. The 2D hexagonal Brillouin zones for the nitrogen-doped polyarylene networks (NPNs) and the nanoribbons (NPCNs) were sampled using $16 \times 16 \times 1$ and $24 \times 1 \times 1$ Monkhorst-Pack grids, respectively[53]. The convergence thresholds of the total energy and atomic force were set to $10^{-4}$ eV and 0.02 eV/Å, respectively.

## Data availability

The data that support the findings of this study are available from the corresponding authors upon reasonable request.

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

## Acknowledgements

This work was financially supported by the NSFC Grant No. 21972101.

## Author contributions

ZS and JH conceived and designed the project. JX and SX performed STM experiments and data analysis. JH performed DFT calculations. All authors discussed the results and contributed in the manuscript preparation.

## Competing interests

The authors declare no competing interests.
