## [Peer Review File · Communications Chemistry]

Reviewers' comments:

Reviewer #1 (Remarks to the Author):

In the manuscript, the authors reported the on-surface synthesis of nitrogen-doped porous carbon nanoribbons (NPCNs) on metal surfaces, by the intrinsic selectivity in the molecular configuration and a two-step reaction pathway. The authors also examined the electronic properties of the resulting carbon nanoribbons by using DFT calculations. The manuscript is well written, the STM and DFT results are well presented, and the data analysis and discussion support the main conclusion. I would suggest the acceptance in Communications Chemistry upon minor revisions be addressed as listed:

1. The results on Cu(111) were much different from those on Au(111) and Ag(111) (as shown in Figure 4 and Figure S3). The authors attributed this difference to the substrate catalytic property and Cu-coordination (on Cu(111)). Many literatures documented the existence of organometallic intermediates (C-Ag or C-Cu) for on-surface Ullmann reactions. Did authors observe any organometallic intermediates, prior to the formation of ZZC, for the systems on Ag(111) and Cu(111) surfaces? Did the intermediates (if there were) affect the selectivity of reaction pathways?
2. In Figure S1 and S3, the author showed the length distributions. Please also give the sample size for the corresponding distributions.
3. On Au(111), were there any nanoribbons wider than the 6-NPCN? Why the n-NPCNs on Cu(111) was much narrower than that for Au(111) and Ag(111)? The author should explain or give comments on this width limit.
4. I suggest the authors to calculate the PDOS for freestanding NPCNs, which may offer insights into the electronic properties of the ribbons.
5. Some typos should be revised carefully.

Reviewer #2 (Remarks to the Author):

The authors report on the on-surface synthesis of a new type of N-doped porous graphene nanostructure. The synthesis is characterized by a sequential strategy that goes from the formation of zigzag chain units to their progressive coupling that leads to nanoporous ribbons of different width. The STM analysis of each reaction step allows the description of the synthetic details, and the systematic study of different substrates evidence the important role of the substrate. Finally, they provide ab-initio calculations of the band structure of the observed products, to conclude that they are all semiconducting, with the gap depending on the width of the ribbon, as expected from quantum

confinement effects. The results do not reveal any new synthetic methodology nor any unexpected physical properties of the products. However, the main challenge for the progress in the field of onsurface synthesis of carbon-based nanomaterials is the introduction of heteroatoms and the formation

of complex structures that go beyond the continuous (non-porous) graphene nanoribbons. As such, I consider the results relevant enough for publication in this journal, subjected, however, to clarifying some questions and answering to some comments that I list below:

1. The introduction is focused on the on-surface synthesis of carbon-based nanoarchitectures, but the state of art in the synthesis of nitrogen-doped and nanopatterned nanostructures is too narrow. See for instance the 2D nanoporous structures that have been reported previously (non of them cited here): M. Bieri et al., *Chem. Comm.* 45 6919 (2009); C. Moreno et al. *Science* 360, 199 (2018); C. Moreno et al. *J. Am. Chem. Soc.* 145, 16, 8988 (2023); D. Wang et al., *ACS Nano* 14, 14008 (2020); M. Tenorio et al. *Adv. Mater.* 34, 2110099 (2022).

All these are pioneering examples of nanopatterned or porous graphene nanostructures, and the last two are N-doped. As such, I missed them in the introduction.

2. Inter-chain cyclodehydrogenation: Since the inter-chain coupling does not lead to any cyclization or ring closure, I think it has to be referred to as de-hydrogenative cross-coupling, not cyclodehydrogenation.

3. Br-mediated interchain interactions: a very similar self-assembly has been previously reported for the undoped polyarylene counterpart, it should be cited then, and any differences/similarities discussed: I. Piquero-Zulaica et al., *ACS Nano* 12, 10537 (2018). Also, the precursor self-assembly has been explained in terms of Br-Br interactions, but similar BrH as the ones mediating the interchain interactions could exist according to the schematic drawn in Fig. 2b. The authors should provide more evidence for their conclusion or leave more open scenarios.

4. Figure 3: the mobility of individual chains inferred from their appearance in Fig. 3a cannot be judged due to the low resolution of the images (extendable to all figures).

5. When discussing the activation temperature of interchain coupling in Ag(111), it is said that it agrees fairly to the case of Au(111), but it does not need to be like that given the different reactivity of Au(111). So the sentence of pg. 6: "Such an onset temperature agrees fairly with

the activation temperature (635–650 K) for the cyclodehydrogenation on Au(111)” does not have any scientific foundation. Also, I don’t agree with the following sentence, “The two-step reactions on Ag(111) and Au(111) showed similar behaviors, in agreement with previous reports.” The yield in the two cases is radically different, why is it? The differences should be discussed.

6. For the control experiment with Cu, authors deposit Cu atoms on Au. From the observation of individual rings in this system they conclude that Cu coordination is the origin for such structures. But in the Fig. S2 one can see that Cu atoms rearrange in islands, and the observed rings are on top of what it seems a Cu island. The authors should comment on that and provide further evidence for the coordination scenario.

7. In Methods, a sample temperature range of 113-293K is reported. Temperature for each image should be instead provided, specially to understand if the mobile chains are obtained at the highest temperature or a even mobile at 100 K.

Reviewer #3 (Remarks to the Author):

In the manuscript entitled “Stepwise Synthesis of Nitrogen-Doped Porous Carbon Nanoribbons on Surface”, J. Xu et al. studied on-surface reactions of 6,6''-dibromo-2,2':6',2''-terpyridine (DT) molecule on Au, Ag and Cu surfaces, synthesizing polymer chains and 2D porous nanoribbons with different widths. STM imaging provided the solid evidence of the formation of these products; and DFT calculations have revealed the electronic properties of the synthesized structures. The data demonstrated in the manuscript has fully supported the conclusions. However, there still leaves some other important open questions that authors should take into account before publication, which are described in the following:

- There are many previous studies announced that, when molecules like DT reacting, organometallic intermediates forms before metal-free products on Ag(111) and Cu(111) surfaces. The data of organometallic intermediates is missing in the current version.
- The authors performed control experiments by adding DT molecules to the Au(111) surface preadsorbed with Cu atoms. Similar to the last comment, I wonder whether the authors experimentally observed the formation of organometallic intermediates.
- In Figure S2b, are polymers adsorbed on Cu island/Au(111)? The authors should add more detailed description to Figure S2.
- What are the experimental results when the coverage is fairly low on Au(111), oligomer chains or rings?

- As for HSE06 results, why just show $K \rightarrow \Gamma \rightarrow M'$ parts?

- The authors should illustrate that the calculated electronic structures are just for freestanding n-NPCNs, without surfaces.

This study is therefore suitable for publication in Communications Chemistry after minor revisions.

Response to Reviewers' Comments

We thank the reviewers for their reading and suggestion on our manuscript. According to the comments, we have revised the manuscript. All revisions were highlighted in yellow. Our point-by-point response is as follows:

Reviewer #1 (Remarks to the Author):

In the manuscript, the authors reported the on-surface synthesis of nitrogen-doped porous carbon nanoribbons (NPCNs) on metal surfaces, by the intrinsic selectivity in the molecular configuration and a two-step reaction pathway. The authors also examined the electronic properties of the resulting carbon nanoribbons by using DFT calculations. The manuscript is well written, the STM and DFT results are well presented, and the data analysis and discussion support the main conclusion. I would suggest the acceptance in Communications Chemistry upon minor revisions be addressed as listed:

1. The results on Cu(111) were much different from those on Au(111) and Ag(111) (as shown in Figure 4 and Figure S3). The authors attributed this difference to the substrate catalytic property and Cu-coordination (on Cu(111)). Many literatures documented the existence of organometallic intermediates (C-Ag or C-Cu) for on-surface Ullmann reactions. Did authors observe any organometallic intermediates, prior to the formation of ZZC, for the systems on Ag(111) and Cu(111) surfaces? Did the intermediates (if there were) affect the selectivity of reaction pathways?

We did not observe the organometallic intermediates for the Ag(111) and Cu(111) sample. This result agrees with our previous work [Liang, H., et al, *ChemPhysChem* **2020**, *21* (9), 843-846]. We do not exclude the presence of the intermediates for the two samples, which probably appear and only be stable at lower temperatures than that applied in our experimental conditions.

2. In Figure S1 and S3, the author showed the length distributions. Please also give the sample size for the corresponding distributions.

We have provided the information in the revised manuscript; see the revised supplementary note corresponding to Figure S1 and S4.

3. On Au(111), were there any nanoribbons wider than the 6-NPCN? Why the n-NPCNs on Cu(111) was much narrower than that for Au(111) and Ag(111)? The author should explain or give comments on this width limit.

On Au(111), we did not observe the nanoribbons wider than the 6-NPCN. About the width limit, according to our scenario proposed, wider nanoribbons can form through multiple steps of dehydrogenative fusion ($\text{ZZC} + (n-1)\text{-NPCN} \rightarrow n\text{-NPCN}$), in which the reaction precursors (ZZCs and (n-1)-NPCNs) size determines the size of the products (n-NPCNs). Thus, the width (and length) limit of NPCNs on Cu(111) can be attributed to the presence of non-chain products and ZZCs having short length after the debromination reaction of DT. To make the point clear, we have revised the main text on this point; see line 19-23, page 7.

4. I suggest the authors to calculate the PDOS for freestanding NPCNs, which may offer insights into the electronic properties of the ribbons.

We thank the reviewer's suggestion. We have conducted the calculation and discussed the electronic properties in detail. See added figures 5b, 6a and S5, and the revised main text in line 22-32, page 8; line 9-17, page 9.

5. Some typos should be revised carefully.

We have double checked thoroughly the manuscript. All typos have been corrected. See 'there' in line 8, 'gives' in line 19, page 3; 'of' in line 14, page 5; 'had' in line 8, page 7.

Reviewer #2 (Remarks to the Author):

see attached report

The authors report on the on-surface synthesis of a new type of N-doped porous graphene nanostructure. The synthesis is characterized by a sequential strategy that goes from the formation of zigzag chain units to their progressive coupling that leads to nanoporous ribbons of different width. The STM analysis of each reaction step allows the description of the synthetic details, and the systematic study of different substrates evidence the important role of the substrate. Finally, they provide ab-initio calculations of the band structure of the observed products, to conclude that they are all semiconducting, with the gap depending on the width of the ribbon, as expected from quantum confinement effects. The results do not reveal any new synthetic methodology nor any unexpected physical properties of the products. However, the main challenge for the progress in the field of on-surface synthesis of carbon-based nanomaterials is the introduction of heteroatoms and the formation of complex structures that go beyond the continuous (non-porous) graphene nanoribbons. As such, I consider the results relevant enough for publication in this journal, subjected, however, to clarifying some questions and answering to some comments that I list below:

1. The introduction is focused on the on-surface synthesis of carbon-based nanoarchitectures, but the state of art in the synthesis of nitrogen-doped and nanopatterned nanostructures is too narrow. See for instance the 2D nanoporous structures that have been reported previously (non of them cited here): M. Bieri et al., Chem. Comm. 45 6919 (2009); C. Moreno et al. Science 360, 199 (2018); C. Moreno et al. J. Am. Chem. Soc. 145, 16, 8988 (2023); D. Wang et al., ACS Nano 14, 14008 (2020); M. Tenorio et al. Adv. Mater. 34, 2110099 (2022). All these are pioneering examples of nanopatterned or porous graphene nanostructures, and the last two are N-doped. As such, I missed them in the introduction.

We thank the reviewer for her/his suggestion. We have revised the paper's introduction, and included more relevant references, which were cited to introduce particularly the on-surface synthesis of 2D nanopatterned carbon structures; see revisions in main text in line 25-27 and the added citations 23-26 in page 1. The paper (M. Bieri et al., Chem. Comm. 45 6919 (2009)) was included in previous version as Ref. 1.

2. Inter-chain cyclodehydrogenation: Since the inter-chain coupling does not lead to any cyclization or ring closure, I think it has to be referred to as de-hydrogenative cross-coupling, not cyclodehydrogenation.

We agree with the reviewer, and have modified the manuscript; see line 15, page 1; line 11, page 2; line 6 page 3, and etc.

3. Br-mediated interchain interactions: a very similar self-assembly has been previously reported for the undoped polyarylene counterpart, it should be cited then, and any differences/similarities discussed: I. Piquero-Zulaica et al., ACS Nano 12, 10537 (2018). Also, the precursor self-assembly has been explained in terms of Br-Br interactions, but similar Br-H as the ones mediating the interchain interactions could exist according to the schematic drawn in Fig. 2b. The authors should provide more evidence for their conclusion or leave more open scenarios.

According to the reviewer's comments, we have revised the main text on Br-mediated interchain interactions and also cited the reference mentioned. We also measured the Br-H distance in the precursor assembly, which suggested Br-H hydrogen bonds assisting to stabilize the assembly. Because the vary large number of literatures discussed Br-Br and Br-H bonds on surfaces, and particularly very similar results have been reported in a recent work (Patera, L. L. et al, *Chemistry* **2022**, 4 (1), 112-117; the paper was also cited in our manuscript), we do not discuss ZZCs in detail, but focus on the following dehydrogenation fusion of nanoribbons. See revisions in line 2-5, page 4; and the added reference cited as Ref. 35 in revised version.

4. Figure 3: the mobility of individual chains inferred from their appearance in Fig. 3a cannot be judged due to the low resolution of the images (extendable to all figures).

We have added a figure (Figure. S2) to show the motion of individual chains; see Figure S2 in SI. We have also modified some figures by using high-resolution images; see modified figures Fig. 1, 3a, 5a and 6b.

5. When discussing the activation temperature of interchain coupling in Ag(111), it is said that it agrees fairly to the case of Au(111), but it does not need to be like that given the different reactivity of Au(111). So the sentence of pg. 6: "Such an onset temperature agrees fairly with the activation temperature (635–650 K) for the cyclodehydrogenation on Au(111)" does not have any scientific foundation. Also, I don't agree with the following sentence, "The two-step reactions on Ag(111) and Au(111) showed similar behaviors, in agreement with previous reports." The yield in the two cases is radically different, why is it? The differences should be discussed.

We thank the reviewer for pointing this out. As indicated in our experiments, the activation temperature of dehydrogenation on Ag(111) was close to that for Au(111) surfaces. In comparison with the Cu(111) samples, the activation temperature for Ag(111) is quite close to Au(111), which agrees with the trend of on-surface Ullmann-like reactions on these coinage surfaces, and thus reflects a similar reactivity trend for the dehydrogenation. Also, "The two-step reactions ...behaviors, ..." was stated to compare the three samples. In comparison with the Cu(111) samples, the reaction behavior on Ag(111) is much similar to that for Au(111). Accurate description of the activation temperature and catalytic property of different surfaces on dehydrogenative cross-coupling may need real-time XPS measurements, and thus is out of our instrument limit. We have revised the main text to make our point clear; see line 16-20, page 6. The sentence "The two-step reactions ... with previous reports.", as the reviewer thinks, was somehow misleading, and thus was deleted in the revised version.

6. For the control experiment with Cu, authors deposit Cu atoms on Au. From the observation of individual rings in this system they conclude that Cu coordination is the origin for such structures. But in the Fig. S2 one can see that Cu atoms rearrange in islands, and the observed rings are on top of what it seems a Cu island. The authors should comment on that and provide further evidence for the coordination scenario.

We thank the reviewer for pointing this out. The polymers/rings shown in Fig. S2 did be on top of a Cu island. We have replaced the figure using a new STM image. Nevertheless, our conclusion does not change, in that kinked chains and rings are visible on the gold surface. See modified Figure S3 in revised SI.

7. In Methods, a sample temperature range of 113-293K is reported. Temperature for each image should be instead provided, specially to understand if the mobile chains are obtained at the highest temperature or a even mobile at 100 K.

We have provided the information in the revised manuscript. The mobile chains are observed during the scanning at 293 K. See added sentence in in line 22-23 , page 10.

Reviewer #3 (Remarks to the Author):

In the manuscript entitled “Stepwise Synthesis of Nitrogen-Doped Porous Carbon Nanoribbons on Surface”, J. Xu et al. studied on-surface reactions of 6,6"-dibromo-2,2':6',2"-terpyridine (DT) molecule on Au, Ag and Cu surfaces, synthesizing polymer chains and 2D porous nanoribbons with different widths. STM imaging provided the solid evidence of the formation of these products; and DFT calculations have revealed the electronic properties of the synthesized structures. The data demonstrated in the manuscript has fully supported the conclusions. However, there still leaves some other important open questions that authors should take into account before publication, which are described in the following:

- There are many previous studies announced that, when molecules like DT reacting, organometallic intermediates forms before metal-free products on Ag(111) and Cu(111) surfaces. The data of organometallic intermediates is missing in the current version.

We thank the reviewer for her/his suggestion. We did not observe organometallic intermediates on Ag(111) and Cu(111). This result agrees with our previous work [Liang, H., et al, *ChemPhysChem* **2020**, *21* (9), 843-846]. We do not exclude the potential formation of the intermediates for the two samples, which might occur and only be stable at low temperatures out of our experimental conditions.

- The authors performed control experiments by adding DT molecules to the Au(111) surface preadsorbed with Cu atoms. Similar to the last comment, I wonder whether the authors experimentally observed the formation of organometallic intermediates.

We did not observe the organometallic intermediates. Similarly, we propose that the intermediates, if exist, may likely occur out of our experimental conditions.

- In Figure S2b, are polymers adsorbed on Cu island/Au(111)? The authors should add more detailed description to Figure S2.

We thank the reviewer for her/his pointing this out. Yes, we noted that most of polymers shown in Figure S2b adsorbed on an epitaxial Cu island rather than the gold surface. We have used a new image to show the polymers topography; see Figure S3 in revised SI.

- What are the experimental results when the coverage is fairly low on Au(111), oligomer chains or rings?

We prepared a sample with a coverage of DT ~ 0.25 ML on Au(111). As show in Figure A1, after an annealing at 625 K oligomer chains containing many foldings and kinks (and thus shortening in length), rings and 1(2)-NPCNs appeared. Our manuscript focused on the fabrication of nanoribbons, and thus we do not include this image in the revised version.

Figure A1. The STM overview (acquired at room temperature) of a Au(111) sample covered with DTs ~ 0.25 ML. The rings are marked by an oval circle. Data acquisition condition: $U = -1.1$ V, $I = 800$ pA.

- As for HSE06 results, why just show $K \rightarrow \Gamma \rightarrow M'$ parts?

As indicated by our calculation that the band structure has an indirect band gap, $K \rightarrow \Gamma \rightarrow M'$ parts allow us to illustrate both CBM and VBM. We have revised the text to make it clear; see line 19-20, page 8.

- The authors should illustrate that the calculated electronic structures are just for freestanding n-NPCNs, without surfaces.

We have revised the text to make this point clear; see the text in line 6, 7, page 8; and the caption of Figure 5a and 6a.

This study is therefore suitable for publication in *Communications Chemistry* after minor revisions.

REVIEWERS' COMMENTS:

Reviewer #1 (Remarks to the Author):

After a careful reading of this revision, it is apparent that the authors have addressed all concerns and thus I am now in favor of its publication as it stands.

Reviewer #2 (Remarks to the Author):

I consider the authors have addressed all comments satisfactorily and therefore suggest the manuscript for publication.

Reviewer #3 (Remarks to the Author):

In the revised manuscript, the authors have fully considered the referees' reports. Therefore, I recommend this paper to be published in Communications Chemistry.